# Photobiocatalytic synthesis of chiral secondary fatty alcohols from renewable unsaturated fatty acids

Wuyuan Zhang [1,2,6], Jeong-Hoo Lee[3,6], Sabry H. H. Younes[1,4], Fabio Tonin[1], Peter-Leon Hagedoorn [1], Harald Pichler [5], Yoonjin Baeg[3], Jin-Byung Park[3✉], Robert Kourist [5✉] & Frank Hollmann [1✉]

En route to a bio-based chemical industry, the conversion of fatty acids into building blocks is of particular interest. Enzymatic routes, occurring under mild conditions and excelling by intrinsic selectivity, are particularly attractive. Here we report photoenzymatic cascade reactions to transform unsaturated fatty acids into enantiomerically pure secondary fatty alcohols. In a first step the C=C-double bond is stereoselectively hydrated using oleate hydratases from *Lactobacillus reuteri* or *Stenotrophomonas maltophilia*. Also, dihydroxylation mediated by the 5,8-diol synthase from *Aspergillus nidulans* is demonstrated. The second step comprises decarboxylation of the intermediate hydroxy acids by the photoactivated decarboxylase from *Chlorella variabilis* NC64A. A broad range of (poly)unsaturated fatty acids can be transformed into enantiomerically pure fatty alcohols in a simple one-pot approach.

[1] Department of Biotechnology, Delft University of Technology, Van der Maasweg 9, 2629 HZ Delft, The Netherlands. [2] School of Chemical Engineering and Technology, Xi'an Jiaotong University, 710049 Xi'an, China. [3] Department of Food Science & Engineering, Ewha Womans University, Seoul 03760, Republic of Korea. [4] Chemistry Department, Faculty of Science, Sohag University, Sohag 82524, Egypt. [5] Institute of Molecular Biotechnology, Graz University of Technology, Petersgasse 14, 8010 Graz, Austria. [6] These authors contributed equally: Wuyuan Zhang, Jeong-Hoo Lee. ✉email: jbpark06@ewha.ac.kr; kourist@tugraz.at; f.hollmann@tudelft.nl

Envisioning a biobased chemical industry, there is an increasing interest in the transformation of biomass-derived starting materials into chemical building blocks[1,2]. Natural fatty acids are particularly interesting building blocks, especially if derived from agricultural wastes or non-edible sources. Until recently, chemical methodologies for the conversion of fatty acids or their glycerides have been largely restricted to their (trans) esterification for the production of biodiesel[3] or cosmetic esters[4]. This situation is changing dramatically with various research groups developing new chemistries to valorise fatty acids (Fig. 1).

For example, with the discovery of the fatty acid decarboxylase OleT[5–7] or UndA/B[8], synthesis of terminal alkenes from fatty acids has come into reach[6,9–12] giving access to chemical building blocks[13,14]. Also the hydroxylation of fatty acids using P450 monooxygenases[15], per-oxygenases[16,17] or dioxygenases[18] is receiving increasing attention. The resulting hydroxy acids may be interesting building blocks for biobased and biodegradable polyesters. Oxyfunctionalisation of unsaturated fatty acids can also be achieved via selective water addition to the *cis*-C=C-double bond[19,20] or via allylic hydroperoxidation[21] followed by C–C-bond cleavage[22–27] or isomerisation to diols[28,29]. Also the selective reduction of the carboxylate group to either the alcohol or aldehyde moiety is possible[30,31]. Finally, the chemoenzymatic epoxidation of unsaturated fatty acids exploiting the 'perhydrolase' activity of lipases is worth mentioning[32,33].

Long-chain secondary alcohols, which may be active ingredients in cosmetic formulations[34,35], performance additives in oleochemicals or building blocks in natural product synthesis[36] and for organic photosensitisers[37], are currently not accessible from natural fatty acids. Established synthetic routes almost exclusively build on Grignard-type reactions of halide-derived nucleophiles with aldehydes or formic acid esters[37], thereby necessitating multistep syntheses, leading to racemic products and generating significant amounts of salt wastes.

Recently, a decarboxylase from *Chlorella variabilis* NC64A (*Cv*FAP) has been reported[38], enabling the synthesis of alkanes from fatty acids[39,40] or the kinetic resolution of α-substituted acids[41]. Compared to existing chemical decarboxylation pathways[42], *Cv*FAP appears particularly attractive due to the high chemoselectivity of the *Cv*FAP-reaction under mild reaction condition and its high-functional group tolerance (leaving C=C-double bonds and OH-groups present in the starting material unaltered). *Cv*FAP is a photoenzyme, i.e., its catalytic activity depends on the activation by light. More specifically, only the photoexcited flavin prosthetic group is sufficiently reactive for a single electron-transfer from the enzyme-bound carboxylate and thereby to initiate the decarboxylation reaction[38].

Fascinated by the synthetic possibilities offered by *Cv*FAP we became interested in further elucidating its substrate scope and used it for the synthesis of functionalised alkane products (Scheme 2). We envision starting from unsaturated fatty acids, first introducing the alcohol functionality using either a fatty acid hydratase (Fig. 2, Cascade 1 or a diol synthase (Fig. 2, Cascade 2) followed by *Cv*FAP-catalysed decarboxylation.

## Results

**Design of the photoenzymatic cascades.** The photoactivated carboxylic acid decarboxylase *Cv*FAP was produced by recombinant expression in *Escherichia coli* following established protocols[38] (see Supplementary Methods) and used either as cell-free extracts or in whole cells. For the hydration of unsaturated fatty acids we first chose the oleate hydratase from *Lactobacillus reuteri* (*Lr*OhyA). The synthetic gene encoding *Lr*OhyA (Accession number: WP_109913811) was cloned into a pET28 vector and the enzyme was recombinantly expressed in *E. coli* BL21 (DE3) cells (Supplementary Fig. 1). Lyophilised cells containing *Lr*OhyA were used for further reactions. It is worth mentioning here that empty *E. coli* cells (not containing any of the plasmids mentioned above) exhibited neither hydratase nor decarboxylation activity (Supplementary Fig. 5).

We first drew our attention to the hydratase/decarboxylase cascade, which indeed proceeded as envisioned. *Lr*OhyA catalysed the hydration of oleic acid (Supplementary Fig. 6) followed by *Cv*FAP-catalysed decarboxylation of the intermediate

**Fig. 1 Natural fatty acids as building blocks. In recent years, biocatalytic methodologies for the transformation of fatty acids have practically exploded.** For example: **a** hydrolase-catalysed esterification of amidation[4], **b** reductase-catalysed reduction of the carboxylate group to the corresponding aldehyde and alcohol[30,31], **c** P450-peroxygenase-catalysed oxidative decarboxylation yielding terminal alkenes[5–7], **d** photodecarboxylase-catalysed decarboxylation yielding alkanes[38,39], **e** hydratase-catalysed water addition to C=C-bonds[50], **f** lipoxygenase-catalysed allylic hydroperoxidation[21], **g** use of mono-, di- and per-oxygenases for the terminal hydroxylation and further transformation into acids or amines as polymer building blocks[15–18,51], and **h** multi-enzyme cascades yielding short-chain acids[22-27].

**Fig. 2 Proposed photoenzymatic cascades to transform unsaturated fatty acids into secondary alcohols. a** Cascade 1 comprises the (stereoselective) addition of water to C=C-double bonds catalysed by fatty acid hydratases (FAHs) followed by the decarboxylation mediated by the photoactivated decarboxylase from *Chlorella variabilis* NC64A (*Cv*FAP) generating secondary long-chain alcohols; **b** cascade 2 combines 5,8-diol synthase from *Aspergillus nidulans* (*An*DS) with *Cv*FAP yielding diols.

hydroxy acid to yield 9-heptadecanol (Supplementary Fig. 7). To identify the factors influencing the product formation of the photoenzymatic cascade we further used oleate as model substrate. Using cell-free preparations of *Lr*OhyA gave only low-product formation (0.4 mM of the desired 9-heptadecanol starting from oleic acid). We attribute this to a relative poor stability of *Lr*OhyA under these conditions and therefore focussed using *Lr*OhyA in lyophilised whole cells. It is also worth mentioning here that one-pot one-step procedures (i.e., performing the hydration and the decarboxylation reaction at the same time) predominantly yielded the decarboxylation product of oleic acid ((*Z*)-heptadec-8-ene). Wild-type oleate hydratase requires a carboxylic acid function, which precludes hydration of ((*Z*)-heptadec-8-ene[43]. Therefore, for all further experiments we followed a one-pot two-step procedure, i.e., first performing the hydration reaction followed by the addition of *Cv*FAP to the reaction mixture and illumination to promote the decarboxylation reaction.

Full hydration of 7 mM oleic acid (**1a**) was achieved within 11 h while the subsequent photoenzymatic decarboxylation was considerably faster (Fig. 3). A systematic variation of the reaction parameters (Supplementary Fig. 53) confirmed our initial assumption that *Lr*OhyA represents the limiting factor in the catalytic cascade. Relatively high *Lr*OhyA concentrations (lyophilised cells, 15–20 g L$^{-1}$) were necessary to obtain full conversion of oleic acid into the desired product (**1c**) within the time frame of the experiment.

**Investigating the substrate scope.** Encouraged by this proof-of-concept, we further investigated the substrate scope of the photoenzymatic cascade reaction. A broad range of (poly)unsaturated fatty acids were converted into the corresponding alcohols (Fig. 4 and Supplementary Figs. 8–48). Especially, Δ9-unsaturated fatty acids were converted in acceptable to good yields (24–74%) into the corresponding alcohols. In those cases where poor conversion into the desired alcohols was observed, the hydration step was overall limiting (Supplementary Tables 1 and 2) and the corresponding unsaturated alkenes were the main products. We also investigated the optical purity of the corresponding products. Since commercial standards for most of the products were not available, we performed O-acylation of the alcohol product using (*S*)-(+)-*O*-acetylmandelic acid for NMR analysis to determine their optical purity (Supplementary Fig. 49). Very pleasingly, in

most cases, essentially enantiomerically pure products were obtained.

The cascade using linoleic acid (**4a**) was scaled-up (for details see preparative-scale synthesis in Supplementary Methods). From a semi-preparative transformation, overall 82.5 mg (32.5% isolated yield) of the desired optically pure alcohol (**4c**) was obtained.

At this stage, we identified three major limitations of the current reaction system: (1) poor substrate loadings due to the poor solubility of the lipophilic fatty acid starting materials, (2) low overall reaction rates, especially of the hydration step, and (3) the need for two individual catalyst systems (OhyA and *Cv*FAP).

**Use of two-liquid phase reactions.** To address the solubility issue, we evaluated the well-established two-liquid system wherein a hydrophobic organic phase serves as substrate reservoir and product sink[39]. Given the fact, that the fatty acid substrates of interest are generally obtained from natural trigycerides, this appeared a suitable organic phase (Fig. 5). We evaluated this approach by using triolein as organic phase containing 20 mM oleic acid. In this way, 17.4 mM of 9-heptadecanol was obtained starting from 20 mM of oleic acid dissolved in triolein (87% yield, see Supplementary Table 3). The obvious next step was to extend the cascade by a hydrolase step to enable triglycerides as starting materials (Fig. 5). Again using triolein as organic phase the lipase from *Candida rugosa* (*Cr*Lip) catalysed the hydrolysis of the triglyceride while *Lr*OhyA mediated the hydration of the C=C-double bond. After the illumination of the reaction mixture in the presence of *Cv*FAP, 6.9 mM of 9-heptadecanol was observed in the organic phase. In the current setup (devoid of external pH control), the hydrolysis of triolein was very fast, leading to an acidification of the aqueous layer, as confirmed by a pH paper test. As a consequence, the *Cv*FAP-catalysed decarboxylation slowed down considerably and the intermediate hydroxy acid represented the main product. We expect that higher product concentrations will be possible by controlling the pH of the reaction more stringently[39].

**Co-expression of both enzymes.** Next, we addressed the low productivity issue as well as the need for two individual catalysts by constructing a co-expression system in *E. coli*. Instead of using *Lr*OhyA we used the fatty acid hydratase from *Stenotrophomonas maltophilia* (*Sm*OhyA), which had been reported to exhibit a very promising specific activity of 2.7 U mg$^{-1}$ (refs. [44,45]). A previously optimised *Sm*OhyA expression system in *E. coli* (i.e.,

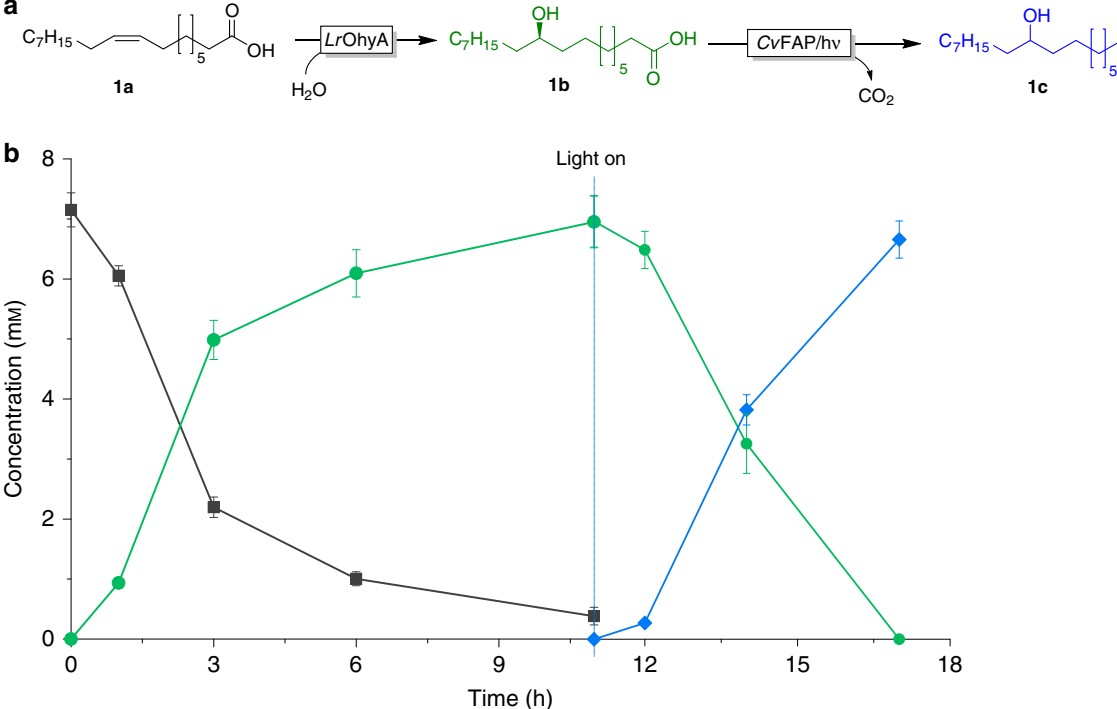

**Fig. 3 Proposed photoenzymatic cascade to transform oleic acid into 9-heptadecanol. a**: Recation scheme. **b** shows a representative time course of the cascade reaction. Reaction conditions: [oleic acid] = 7 mM, [$LrOhyA$ cells] = 15 g L$^{-1}$, [$CvFAP$] = 2 μM, Tris-HCl buffer pH 8.0 (100 mM, with 50 mM of NaCl), illumination with blue light ($\lambda$ = 450 nm; intensity = 13.7 mE L$^{-1}$ s$^{-1}$): oleic acid (black squares), 10-hydroxystearic acid (green circles), 9-heptadecanol (blue diamonds). Values represent the average of duplicates ($n$ = 2). Error bars indicate the standard deviation.

pACYC-PelBSS-OhyA)[46] was used as chassis for the recombinant expression of $CvFAP$ (yielding a recombinant *E. coli* BL21 (DE3) pACYC-PelBSS-OhyA/pET28a-$CvFAP$). Indeed co-overexpression of both enzymes was possible (Supplementary Fig. 4). We, therefore, used this catalyst for the combined hydration/decarboxylation of oleic acid yielding 9-heptadecene (**1c**, Fig. 6).

Despite the lower catalyst loading as compared to the experiment shown in Fig. 3 (7 g$_{CDW}$ L$^{-1}$ instead of 15 g$_{CDW}$ L$^{-1}$) a much higher hydration rate of oleic acid (86 U g$^{-1}$$_{CDW}$) was observed resulting in more than 90% conversion of oleic acid into 10-hydroxyoctadecanoic acid (**1b**) within 7.5 min after which the decarboxylation reaction was initiated by commencing illumination of the reaction mixture with blue light. The rate of the decarboxylation was comparable with the rate shown in Fig. 3. It is worth mentioning that non-converted oleic acid was decarboxylated to (Z)-heptadec-8-ene.

**Enlarging the scope of hydratases**. In addition to the above-used fatty acid hydratases, a range of further fatty acid hydroxylating enzymes (e.g., linoleate 9S-lipoxygenase from *Myxococcus xanthus*[47], 7,10-diol synthase from *Pseudomonas aeruginosa*[29], and 5,8-diol synthase from *Aspergillus nidulans*[28]) have been reported. The 5,8-diol synthase from *A. nidulans* (*An*DS) for example caught our attention as this bifunctional enzyme adds two instead of only one OH functionalities into oleic acid by a two-step reaction (Fig. 7). Thereby, a three step cascade mediated by two enzymes was established for the preparation of (Z)-heptadec-8-ene-4,7-diol (**1e**) from oleic acid.

For the dihydroxylation of oleic acid, 5,8-diol synthase from *A. nidulans* (*An*DS) was used. The first recombinant *E. coli* expressing *An*DS (*E. coli* BL21(DE3) pET21a-*An*DS[48], however, showed only poor *An*DS-activity (Supplementary Fig. 54(A)). Introduction of the signal sequence of PelB directed the enzyme

into the periplasm[46,49]. Notably, *E. coli* BL21(DE3) pACYC-PelBSS-*An*DS displayed approximately 10-fold greater transformation rates and 2.3-fold higher final product concentration, as compared to the original strain *E. coli* BL21(DE3) pET21a-*An*DS (Supplementary Fig. 54(B)). Having a suitable diol synthase and the photodecarboxylase at hand, we performed the conversion of oleic acid (Fig. 7). Already after 1 h, 95% of the starting material had been converted into the diol (**1e**). Initiating the decarboxylation reaction by illumination of the reaction mixture led to an abrupt decrease in all carboxylic acids present to the corresponding alkanes. The chemical identity of the final product as well as the intermediate hydroxy acid were confirmed via GC/MS (Supplementary Fig. 51) and NMR analytics (Supplementary Fig. 50).

Overall, in this contribution we have demonstrated that secondary fatty alcohols can be obtained from unsaturated fatty acids using a cascade of fatty acid hydratase or diol synthase and fatty acid decarboxylase. The substrate scope of the current system is fairly broad giving access to enantiomerically pure alcohols from renewable starting materials. Admittedly, the product titres achieved in this proof-of-concept study are too low to be economically and environmentally attractive. Further work in our groups will focus on the expansion of this proof-of-concept experiments for synthetic application, increasing the product yields and the investigation of their biological properties such as anti-microbial activity.

## Methods

**Preparation of the biocatalysts**. Oleate hydratase from *Lactobacillus reuteri* (*Lr*OH) was produced via recombinant expression in *E. coli* BL21 (DE3) cells harbouring pET28a(+) *Lr*OH (Supplementary Fig. 4). These cultures were grown overnight in lysogeny broth (LB) medium, containing 30 μg mL$^{-1}$ kanamycin. The pre-cultures were used to inoculate large cultures (1000 mL LB + 50 μg mL$^{-1}$ kanamycin in 5 L shake flasks). Cells were grown at 37 °C, 180 rpm, until an OD$_{600}$

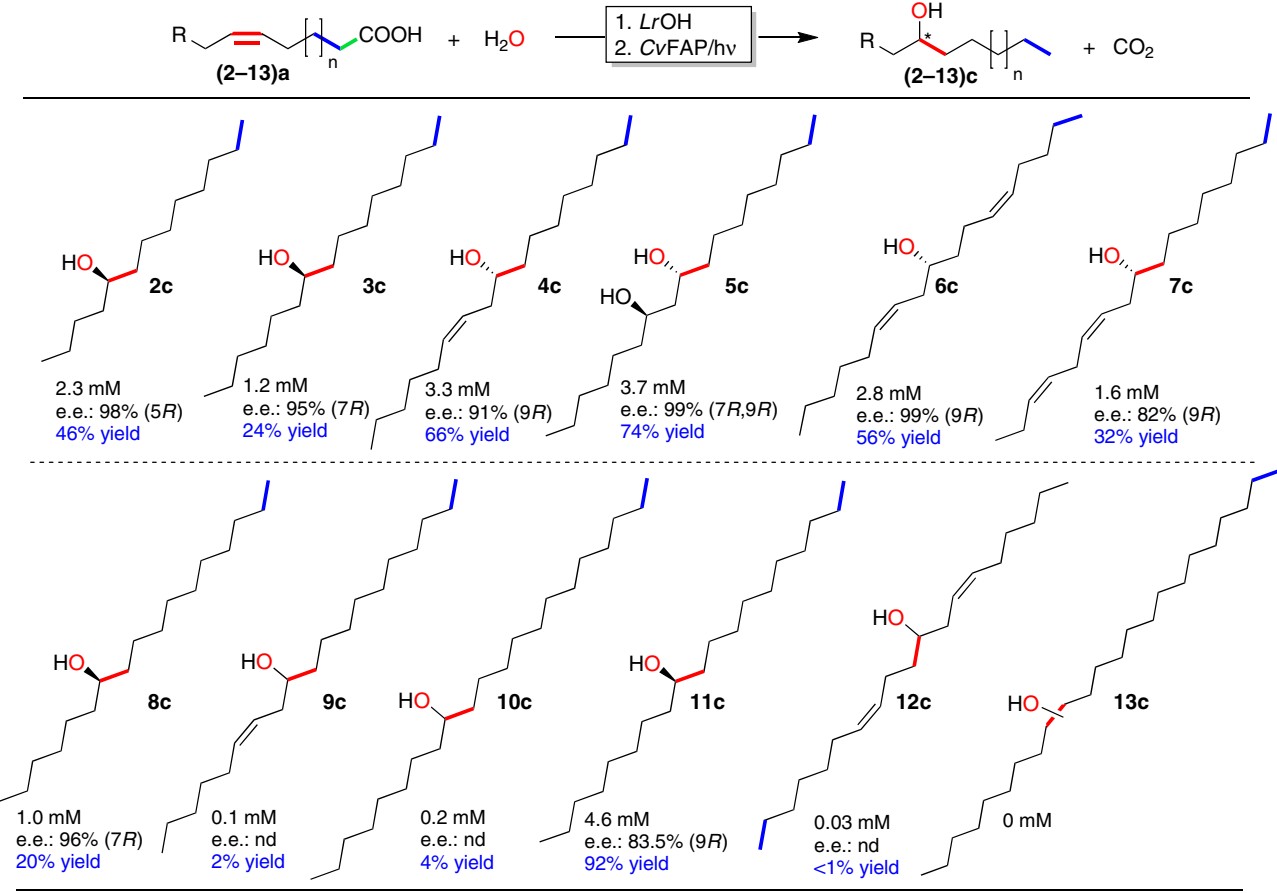

**Fig. 4 Preliminary product scope of the proposed photoenzymatic reaction system.** Reaction conditions: [substrate] = 5 mM, [LrOhyA-cells] = 20 g L$^{-1}$, [$Cv$FAP] = 2 μM, Tris-HCl buffer (100 mM, with 50 mM of NaCl), blue light ($\lambda$ = 450 nm; intensity = 13.7 mE L$^{-1}$ s$^{-1}$). The reactions were performed in a two-step fashion: first the LrOhyA-catalysed hydration reaction was performed for 11 h followed by addition of $Cv$FAP and illumination for another 6 h. nd not determined. Conversion = [product]$_{final}$ × [substrate]$_{initial}^{-1}$ × 100%; determined via GC, conversions determined via $^1$H NMR are shown in Supplementary Tables 2 and 3. The enantiomeric excess (e.e.) was determined by $^1$H NMR analysis after the fatty alcohols were derivatised by ($S$)-( + )-$O$-acetylmandelic acid (details see Supplementary Tables 1 and 2).

between 0.6 and 0.8 was reached. Protein production was induced by the addition of 0.5 mM isopropyl-β-D-thiogalactopyranoside (IPTG) (final concentration) and the cells were left at 20 °C, 180 rpm, for overnight (18 h). Cells were harvested by centrifugation (11,000 × $g$ at 4 °C for 10 min), washed with Tris-HCl buffer (50 mM, pH 7.5, 100 mM NaCl) and centrifuged again. The cell pellets were collected and stored at −80 °C for further use. The expression level of $Lr$OH was found to be rather reproducible (9.7 ± 1 mg $Lr$OH per gram cell dry weight) from various expression experiments at different scales (50 mL to 9.6 L).

5,8-Diol synthase from *Aspergillus nidulans* ($An$DS) was expressed in *E. coli* BL21(DE3) by using the recombinant plasmids (i.e, pET21a-$An$DS[33] and pACYC-PelBSS-$An$DS) (see the SI for details). The recombinant *E. coli* cultures were grown overnight in terrific broth (TB) medium containing the appropriate antibiotics. The pre-cultures were used to inoculate large cultures (500 mL in 2 L shake flasks). The cells were grown at 37 °C, 180 rpm until an OD$_{600}$ between 0.6 and 0.8 was reached. Protein production was induced by the addition of 0.1 mM IPTG and the cells were left at 16 °C, 150 rpm for overnight. The resulting cells were harvested by centrifugation and used as the biocatalysts for dihydroxylation of oleic acid (Fig. 7).

The fatty acid photodecarboxylase from *Chlorella variabilis* NC64A ($Cv$FAP) was produced in *E. coli* BL21 (DE3)[27]. In short, 10 mL pre-cultures of *E. coli* BL21 (DE3) cells harbouring the designed pET28a-His-TrxA-$Cv$FAP plasmid were grown overnight in TB medium, containing 50 μg mL$^{-1}$ kanamycin. From these, 500 mL cultures (TB + 50 μg mL$^{-1}$ kanamycin in 2 L shake flasks) were prepared (cell growth at 37 °C, 180 rpm, until an OD$_{600}$ between 0.7 and 0.8 followed by induction by the addition of 0.5 mM IPTG). The cultures were incubated at 17 °C, 180 rpm, for another 20 h. Cells were harvested (centrifugation at 11,000 × $g$, 4 °C for 10 min) and resuspended directly into the $An$DS reaction medium. Otherwise, the cells, which were harvested (centrifugation at 11,000 × $g$, 4 °C for 10 min), were washed with Tris-HCl buffer (50 mM, pH 8, 100 mM NaCl) and centrifuged again. The cell pellet was suspended in the same buffer, and 1 mM PMSF was added. Cells were lysed by passing them passed twice through a Multi Shot Cell Disruption

System (Constant Systems Ltd, Daventry, UK) at 1.5 bar, followed by centrifugation of the cell lysate (38,000 × $g$ at 4 °C for 1 h). After centrifugation, 5% glycerol (w/v) was added to the soluble fraction, the cell extract was aliquoted, frozen in liquid nitrogen and stored at −80 °C.

The total protein content of the cell extract was determined by a BCA Assay (Interchim), using BSA as a standard. $Cv$FAP production was analysed by sodium dodecyl sulfate–polyacrylamide gel electrophoresis using a Criterion™ Cell electrophoresis system (Bio-Rad).

The recombinant *E. coli* BL21(DE3) pACYC-PelBSS-OhyA/pET28a-$Cv$FAP co-expressing $Sm$OhyA and $Cv$FAP were grown overnight in TB medium, containing appropriate antibiotics. From these, 500 mL cultures (TB + appropriate antibiotics in 2 L shake flasks) were prepared (cell growth at 37 °C, 180 rpm, until an OD$_{600}$ between 0.7 and 0.8 followed by induction by the addition of 0.5 mM IPTG). The cultures were incubated at 20 °C, 180 rpm, for another 20 h. Cells were harvested (centrifugation at 11,000 × $g$, 4 °C for 10 min) and resuspended directly into the Tris-HCl buffer (50 mM, pH 8, 100 mM NaCl) for biotransformation.

**General procedures for cascade reactions.** Experiments were performed as independent duplicates. In all, 2.5–20 mg of lyophilised *E. coli* cells of oleate hydratase, and 2.0 mg of oleic acid were added into 980 μL of Tris-HCl buffer (100 mM, with 50 mM of NaCl) for the hydratase-decarboxylase cascade reaction (Fig. 2). The resultant suspension was stirred at 30 °C for 11 h. 20 μL of photo-decarboxylase (from stock solution with a concentration of 102 μM) was added afterwards and the suspension was illuminated with blue LED light and stirred for another 6 h. The final reaction conditions were: Reaction condition: [substrate] = 7 mM, [lyophilised $Lr$OH cells] = 2.5–20 mg mL$^{-1}$, [$Cv$FAP] = 2 μM, Tris-HCl buffer (pH 8.0, 100 mM, with 50 mM of NaCl), blue light (intensity = 13.7 mE L$^{-1}$ s$^{-1}$), total volume 1.0 mL. To analyse the product, 1 mL of ethyl acetate (containing 5 mM of 1-octanol) was added to the above reaction suspension (1:1 volume ratio) -> 3 mL of ethyl acetate (containing 5 g/L of palmitic acid) was

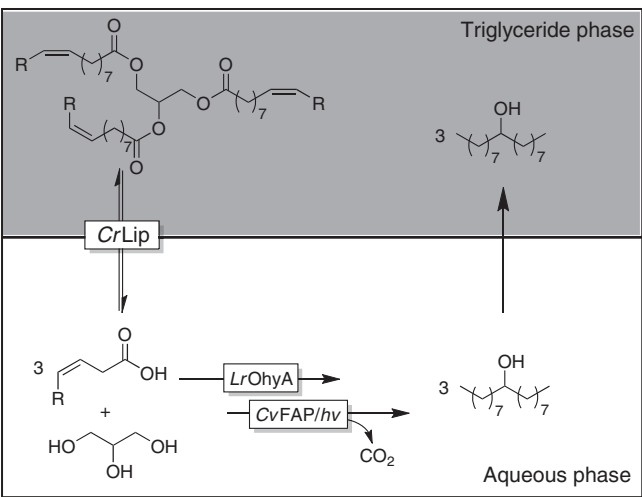

**Fig. 5 Trienzymatic cascade for the transformation of triolein into 9-heptadecanol using a two-liquid-phase approach.** The aqueous reaction medium is supplemented with neat triolein (triglyceride phase) serving as substrate reservoir and product sink. In the reaction sequence, triolein is hydrolysed by the lipase from *Candida rugosa* (*Cr*Lip, located at the interphase) liberating glycerol and oleic acid. The latter is hydrated and decarboxylated (catalysed by *Lr*OHyA and *Cv*FAP) yielding 9-heptadeconol, which partitions back into the hydrophobic phase.

added to the above reaction suspension (3:1 volume ratio) The organic phase was collected by centrifugation and was dried over MgSO$_4$. The obtained sample was analysed by gas chromatography (GC) (Cp sil 5CB, column 50 m × 0.53 mm × 1.0 μm).

For the photoenzymatic *Sm*OhyA-hydration and *Cv*FAP-decarboxylation of oleic acid, recombinant *E. coli* BL21 (DE3) pACYC-PelBSS-OhyA/pET28a-CvFAP was added into 50 mM Tris-HCl buffer (pH 6.5) containing 5 mM oleic acid. For the reaction, first the *Sm*OhyA-catalysed hydration reaction was performed for 0.125 h followed by CvFAP-catalysed decarboxylation under illumination for another 1.625 h. The final reaction conditions were: reaction condition: [oleic acid] = 5 mM, [*E. coli* co-expressing *Sm*OhyA and *Cv*FAP] = 7 g L$^{-1}$, Tris-HCl buffer pH 6.5 (50 mM), illumination with blue light ($\lambda = 450$ nm; intensity = 13.7 mE L$^{-1}$ s$^{-1}$).

For the photoenzymatic diol synthesis-decarboxylation of oleic acid (Fig. 7), 7 mg *E. coli* cells containing 5,8-diol synthase (*An*DS cells) and 7 mg of oleic acid were added into 980 μL of HEPES buffer pH 7.5 (50 mM, with 10% (v/v) DMSO). The resultant suspension was stirred at 40 °C for 2 h. Afterwards, 7 mg *E. coli* cells containing photodecarboxylase (*Cv*FAP cells) was added and the suspension was illuminated with blue LED light and stirred for another 7 h. The final reaction conditions were: [oleic acid] = 15 mM, [*An*DS cells] = 7 g L$^{-1}$, [*Cv*FAP cells] = 7 g L$^{-1}$, HEPES buffer pH 7.5 (50 mM, with 10% (v/v) DMSO), blue light (intensity = 13.7 mE L$^{-1}$ s$^{-1}$), total volume 1 mL. To analyse the product, 3 mL of ethyl acetate (containing 5 g L$^{-1}$ of palmitic acid as internal standard) was added to the above reaction suspension (3:1 volume ratio) and vigorously mixed. The organic phase was collected by centrifugation and was dried over MgSO$_4$. The obtained sample was analysed by gas chromatography/mass spectrometry (GC/MS)[15,35]. The results are included in the Supplementary Fig. 51.

**Preparative-scale synthesis starting from linoleic acid.** 98 millilitres of Tris-HCl buffer (pH 8.0, 100 mM, with 50 mM of NaCl) containing 10 mM of substrate and 2 g of lyophilised *Lr*OH cells were mixed in a beaker and stirred at 30 °C for

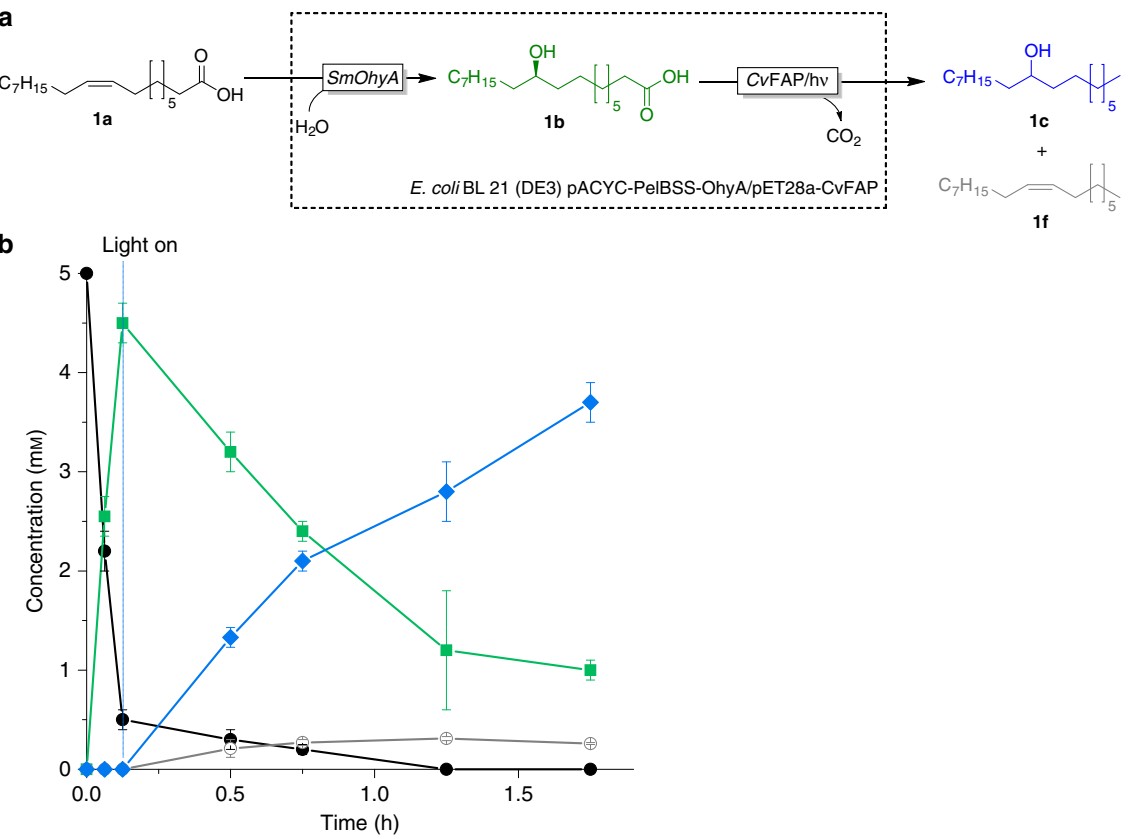

**Fig. 6 Photoenzymatic cascade. a** Reaction scheme of the photoenzymatic cascade combining *Sm*OhA and *Cv*FAP in a single expression host. **b** Time course of the conversion of oleic acid using co-expressed enzymes. Oleic acid (**1a**, black circles) was converted via 10-hydroxystearic acid (**1b**, green squares) into 9-heptadeconol (**1c**, blue diamonds) and the side-product (*Z*)-heptadec-8-ene (**1f**, grey empty circles) using the freshly designed, all-inclusive *E. coli* BL21 (DE3) pACYC-PelBSS-OhyA/pET28a-CvFAP. [oleic acid] = 5 mM, [*E. coli* co-expressing *Sm*OhyA and *Cv*FAP] = 7 g dry cells L$^{-1}$, Tris-HCl buffer pH 6.5 (50 mM), illumination with blue light ($\lambda = 450$ nm; intensity = 13.7 mE L$^{-1}$ s$^{-1}$). For the reaction, first the *Sm*OhyA-catalysed hydration reaction was performed for 0.125 h followed by *Cv*FAP-catalysed decarboxylation under illumination for another 1.625 h. Values represent the average of duplicates (*n* = 2). Error bars indicate the standard deviation.

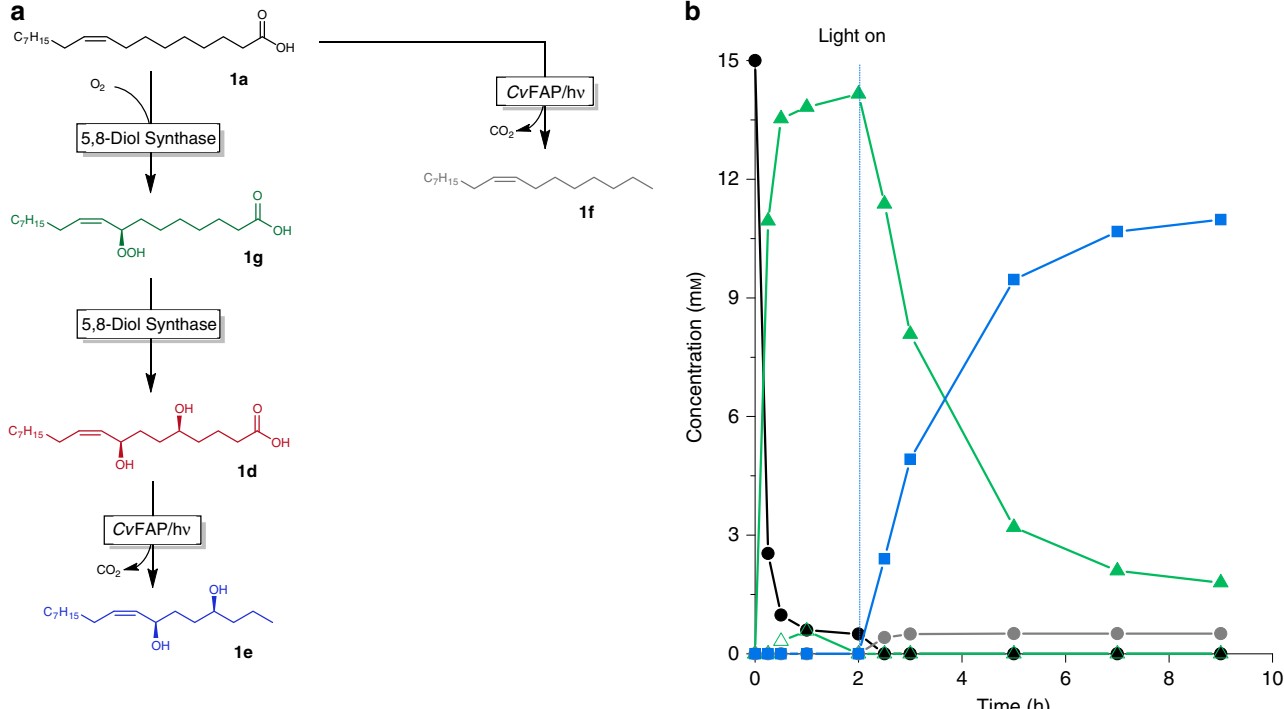

**Fig. 7 Photoenzymatic cascade. a** Reaction scheme of the photoenzymatic cascade transforming oleic acid into Photoenzymatic diol synthesis-decarboxylation of oleic acid. **b** Typical time course [oleic acid] = 15 mM, [AnDS cells] = 7 g L$^{-1}$, [CvFAP cells] = 7 g L$^{-1}$, HEPES buffer pH 7.5 (50 mM, with 10% (v/v) DMSO), illumination with blue light ($\lambda$ = 450 nm; intensity = 13.7 mE L$^{-1}$ s$^{-1}$): oleic acid (black circles), 8-hydroperoxy-9(Z)-octadecenoic acid (**1g**, green empty triangles), 5,8-dihydroxy-9(Z)-octadecenoic acid (**1d**, green triangles), (4S,7R,Z)-heptadec-8-ene-4,7-diol (**1e**, blue squares), (Z)-heptadec-8-ene (**1f**, grey circles). For the reaction, first the AnDS-catalysed diol synthetic reaction was performed for 2 h followed by addition of CvFAP and illumination for another 7 h. The absolute configuration is based on the enantioselectivity of the synthase as previously established by Oh and coworkers[28,48].

48 h. The beaker was sealed by using parafilm. Two millilitres of photo-decarboxylase (from stock solution with a concentration of 102 μM) was added afterwards and the suspension was illuminated by blue LED and stirred for 48 h. The final reaction condition was: [linoleic acid] = 10 mM, [lyophilised LrOH cells] = 20 mg mL$^{-1}$, [CvFAP] = 2 μM, Tris-HCl buffer (pH 8.0, 100 mM, with 50 mM of NaCl), blue light (intensity = 13.7 mE L$^{-1}$ s$^{-1}$), total volume 1.0 mL. At the end of the cascade reactions, the mixture was extracted with ethyl acetate (75 mL, 2×). The extraction solvent of the combined phases was removed under reduced pressure. The crude product was purified via flash chromatography (liquid loading) on silica gel using heptane/ethyl acetate 40:1 as eluent for 15 min, followed by a programmed gradient for 10 min (ethyl acetate/heptane (2.5 to 80% ethyl acetate/heptane gradient). 82.5 mg (32.5% isolated yield) of the corresponding alcohol was obtained starting from linoleic acid.

**Reporting summary.** Further information on research design is available in the Nature Research Reporting Summary linked to this article.

## Data availability

The data that support the findings of this study are available from the corresponding authors upon reasonable request. The source data underlying Figs. 3, 6, 8 and Supplementary Figs. 7 and 25 are provided as a Source data file.

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

## Acknowledgements

The Netherlands Organisation for Scientific Research (NWO) is gratefully acknowledged for financial support through a VICI grant (no. 724.014.003). W.Z. gratefully acknowledges financial support by "Young Talent Support Plan" of Xi'an Jiaotong University (No. 7121191208). This work was also supported by the National Research Foundation of Korea (NRF) grant funded by the Korea government (MEST) (No. 2020R1A2B5B03002376).

## Author contributions

J.B.P., R.K. and F.H. conceived the study, supervised the experimental work and data analysis and wrote the manuscript. W.Z., F.T., Y.B., S.H.H.Y. and J.H.L. performed the reactions and data collection. H.P. and P.L.H. conceptually contributed to the study design. All authors were involved in the composition of the manuscript.

## Competing interests

The authors declare no competing interests.
