## [Peer Review File · Nature Communications]

Reviewers' comments:

Reviewer #1 (Remarks to the Author):

This manuscript entitled "Photobiocatalytic synthesis of chiral secondary fatty alcohols from renewable unsaturated fatty acids" by Hollmann and co-workers described a new enzyme cascade system comprising a alcohol-generating enzyme and a photodecarboxylase to produce enantiomerically pure secondary fatty alcohols from a number of unsaturated fatty acids. Moreover, some preliminary optimization for reaction conditions, cascade extension by inclusion of a lipase, and process engineering were also conducted. The cascade design is novel and interesting. However, there exist some significant problems in this work.

1. The introduction of multi-enzyme cascades starting from fatty acids is not enough. The background information for FAP, LrOH and AnDS is too limited.
2. In Scheme 1, apparently, there are more synthetic routes for production of secondary fatty alcohols. For example, direct hydroxylation of alkanes or alkenes by peroxides, peroxygenases, monooxygenases, or dioxygenases.
3. The results shown in Figure 1 cannot justify one of the major conclusions as stated in line 122 - "All alcohol products were essentially optically pure...". There is one ee% that is only 83.5%, not to mention that the majority of ee% values were not determined. Moreover, the varying yields in Figure 1 are worth more studies and detailed analysis.
4. The amounts of LrOH in *E. coli* should be quantified considering that the expression levels could vary dramatically.
5. In line 173, the authors stated that "co-expression of both enzymes aiming at a 'all inclusive' catalyst system is ongoing". This should be done in this work.
6. The whole manuscript is poorly organized. There are so many one-sentence paragraphs and the logic is difficult to follow. The language also needs to be significantly improved. Some typos (e.g. "stating" in line 73) should be corrected.

Overall, significant improvements have to be made for this work in order to meet the high standard of Nature Communications.

Reviewer #2 (Remarks to the Author):

In this work, Prof. Hollmann Prof. Kourist and Park developed a new one-pot cascade process combining the fatty acid hydratase or diol synthase-catalyzed hydroxylation and the fatty acid decarboxylase-catalyzed decarboxylation for the synthesis of chiral secondary fatty alcohols from from renewable unsaturated fatty acids. Two sequential steps involving fatty acid hydratase and fatty acid hydroxylating enzymes were optimized respectively, providing satisfied results for the model reaction. Moreover the substrate scope of the current system is also fairly broad giving access to enantiomerically pure alcohols. This photobiocatalytic synthesis route represents a more practical and environmentally less demanding alternative for the common chemical synthesis of long chain secondary alcohols with highly optical purity. This exciting work is therefore of significant interests to organic chemists and biotechnology engineers and is suitable for publication in Nature Communications with minor revisions.

1. References should be needed in L86-87, P4, for the pH property of CvFAP and LrOH, or some supplementary data of their pH property.
2. It is very interesting to use triolein as the starting material and involve three enzymes in this one pot as shown in Scheme 2. It would be helpful for readers to know clearly the whole reaction process and the changing concentrations of some main intermediate during the reaction time course.
3. References should be needed in L122-123, P6, for the reported stereospecificity of oleate hydratases.
4. The number of Figure shown in L124, P7 is wrong. It should be Figure 2. I suggest the author to add compound numbers for all products or substrates. Thus it will be easy for readers to catch the compound structure, for example in L131, P7, linoleic acid and so on.

5. Moreover, it would be better to add some discussion about the effect of the substrate structures on the reaction results shown in Figure 1 (P7). Concerning the determination of the ee values of these products, I suggest the author to add some explanation in the text for the reason why NMR of the derivatization of (S)-(+)-O-acetylmandelic acid was used instead of chiral GC or HPLC.

6. In the Authors contributions, XX and YJP did not appear in the author list.

7. In Supplementary Information, L273, P12, one molecular formula of TMS should be reversed. In Figure S10, is palmitic acid as the starting substrate right? In Figure S16, the molecular formula is lost.

Overall, the photobiocatalytic synthetic route of enantiomerically pure long chain secondary alcohols is original, the work is technically well-executed and the conclusions are firmly-supported by experimentation. It is therefore suitable for publication in Nature Communications after these comments are addressed.

Reviewer #3 (Remarks to the Author):

Zhang and colleagues report the use of tandem reactions comprised of two enzyme catalysts (hydratases and flavin-dependent decarboxylase) to facilitate the conversion of oleic acid to secondary alcohols as mono or diols. As the authors state, the transformation could potentially be of interest as an alternative and renewable means to access the synthesis of stereo-specific alcohols. The manuscript is primarily comprised of turnover data and some reaction modification (pH and solvent compatibility). Although the majority of work is generally fine, I do not believe it to represent a major advance, nor is the significance sufficient enough for publication in Nature Communications. Ultimately, the reported substrate/product scope is relatively narrow, the products reported are not particularly interesting/valuable, and there is no fundamentally new enzymology established for the enzymes in question. Additionally, as the manuscript is written, it is unclear if the authors are really achieving the significant TTONs that would enable a new synthetic scheme that is more advantageous than current methods. Although the characterization of the products is generally sound (relying on MS and NMR in some cases), the absence of any raw GC data prohibits the description of any potential side or intermediate products, TTONs, and accurate quantitation. Thus, there are several claims in the manuscript with regards to the performance of one or both enzymes as a function of substrate, pH, etc., that are impossible to substantiate. The authors should provide raw GC chromatograms of reactants and products to show conversion of each species to validate several of these claims. A minor point is that some of the authors' product quantitation relies on assumed rather than empirically determined peak response relationships, which would limit accuracy of the reported conversion yields.

Although this work is of some interest, I do not believe that it meets the high bar required for publication of Nature Communications and would be better suited to another journal.

Major points:

It is unclear whether any of the transformations under examination reach completion. What is the yield of the final product. TTON? Side products? No raw GC traces are shown, making it impossible to trace this information with the protocol as given in the SI as the stock concentrations are not indicated.

Line 15 “novel ...hydratase” Why is this novel? See next comment

Line 62: Rationale or literature precedent for this choice?

Line 67: Control data should be shown in the SI

Line 70: “attribute this to a relative poor stability of *Lr*OH under these conditions” Literature precedent or a control reaction? Needs better explanation.

Line 73: stating

Line 74: I do not understand the nomenclature with @. It seems awkward

Line 76: “predominantly yielded the decarboxylation product of oleic acid” This implies that *Lr*OH cannot dehydrate an alkene. Is there precedent in the hydratase literature? This should be tested directly to demonstrate the substrate scope of *Lr*OH

Line 81: “considerably faster....”

Line 85 “pH has a significant influence ...both enzymes” Unclear from the figure as to whether the pH is affecting one or both enzymes as only the final product (or consumption of the initial substrate) is being followed here.

Line 99 “determination of hydratase activitynot straight forward (sic)” vague. I do not know what the authors are referring to.

Figure 1 legend: What does the quantity for @E. coli refer to? LrOH? Total protein? The text indicates that this is total “lyophilized cells”. However, this implies otherwise

Line 104 “substrate reservoir...product sink” Is this known that the alcohol products will partition in the organic layer as shown? It would also be worth comparing inclusion/omission of the cosolvent system to determine how/if this enables better conversion.

Line 108 “quasi-irreversible hydratation” Why quasi?

Line 110 – “acidification of aqueous layer” as the authors have a simple solution in mind, it would be rather easy to test in a systematic fashion

Line 111-114 Neither the acidification of the aqueous layer nor the “decarboxylation slowed down” are shown. What products are observed in this setup?

Line 119 there is now Figure 2, but rather two Figure 1.

Line 123 “in line with the reported...” a reference is needed. Is this particular ortholog known to be stereospecific? Any other lines of evidence?

137 – “Especially 5,8-diol synthase from ...caught our attention” I do not really understand the impact of this choice. Is (Z)-heptadec-8-ene-4,7-diol particularly valuable?

Line 145 “which we attribute” Completely unsubstantiated comment regarding the cytotoxicity. There are a myriad of other potential reasons the authors may wish to consider.

Line 162, Figure 3: These products should be shown. Where does the hydroxyperoxide derive from? The synthetic reaction was performed for “m h”?

SI line 213 “The derivatized 213 fatty alcohols and 9-heptadecanol were assumed to have the same GC response factor.” This is very unlikely to be correct and casts some doubts as to the reported yields

Minor points

Overall, I found there to be significant problems with the accuracy and quality of the writing. I would also give some thought as to what constitutes a paragraph, and several instances of absent/incorrect citation.

line 24: improper use of a semicolon

line 40: verb incorrect tense

line 40: “broadening its substrate scope” I don’t understand how the enzyme substrate scope has been significantly broadened in the present work beyond the addition of an alcohol.

line 69: incorrect use of colon

Line 105: “See SI” Where exactly?

Detailed answer to the reviewers

Reviewer #1

This manuscript entitled "Photobiocatalytic synthesis of chiral secondary fatty alcohols from renewable unsaturated fatty acids" by Hollmann and co-workers described a new enzyme cascade system comprising a alcohol-generating enzyme and a photodecarboxylase to produce enantiomerically pure secondary fatty alcohols from a number of unsaturated fatty acids. Moreover, some preliminary optimization for reaction conditions, cascade extension by inclusion of a lipase, and process engineering were also conducted. The cascade design is novel and interesting. However, there exist some significant problems in this work.

1. The introduction of multi-enzyme cascades starting from fatty acids is not enough. The background information for FAP, LrOH and AnDS is too limited.

2. In Scheme 1, apparently, there are more synthetic routes for production of secondary fatty alcohols. For example, direct hydroxylation of alkanes or alkenes by peroxides, peroxygenases, monooxygenases, or dioxygenases.

3. The results shown in Figure 1 cannot justify one of the major conclusions as stated in line 122 - "All alcohol products were essentially optically pure...". There is one ee% that is only 83.5%, not to mention that the majority of ee% values were not determined. Moreover, the varying yields in Figure 1 are worth more studies and detailed analysis.

Thank you very much for sharing our enthusiasm for this novel cascade and the synthetic possibilities it brings!
We have done our utmost best to address your comments and suggestions.

Following the reviewer's suggestion, we have improved the introductory section also paying more attention to the enzymes used in this study.

The referee is certainly right with her/his comment that other enzymatic systems exist for the hydroxylation of alkanes. We have added a passage discussing these enzymes and have updated Scheme 1.
It is, however, worth mentioning that most of these methods yield ω , ω -1 and ω -2 alcohols and thereby are not (yet) feasible for the synthesis of (internal) alcohols. Oleate hydratase has the additional advantage that this enzyme class has already applied industrially (for hydroxy acid production for cosmetics), which demonstrates that these enzymes have the necessary stability and activity for synthetic use. We have explained the choice of the model enzymes in the introduction.

We apologise for not having stated the optical purities more clearly in the original manuscript. In addition, we have performed more preparative scale reactions establishing the optical purity of all products obtained in significant amounts. In summary, only two examples have medium optical purities in the range of 80%;

all other products were >90% ee.

4. The amounts of LrOH in E. coli should be quantified considering that the expression levels could vary dramatically.

Following the reviewer's suggestion we have quantified the expression level of LrOH in E. coli. Actually, the expression level turned out to be quite reproducible in various independent experiments. The results are now included in the revised manuscript.

5. In line 173, the authors stated that "co-expression of both enzymes aiming at a 'all inclusive' catalyst system is ongoing". This should be done in this work.

Following the reviewer's suggestion we have performed a range of co-expression experiments. We used this opportunity to also switch to a more active fatty acid hydratase generating a more-future-pointing catalyst system.

6. The whole manuscript is poorly organized. There are so many one-sentence paragraphs and the logic is difficult to follow. The language also needs to be significantly improved. Some typos (e.g. "stating" in line 73) should be corrected.

We have done our utmost best to improve the readability of the revised manuscript!

Overall, significant improvements have to be made for this work in order to meet the high standard of Nature Communications.

We hope that the changes made to the original manuscript meet the reviewer's demands.

Reviewer #2

In this work, Prof. Hollmann Prof. Kourist and Park developed a new one-pot cascade process combining the fatty acid hydratase or diol synthase-catalyzed hydroxylation and the fatty acid decarboxylase-catalyzed decarboxylation for the synthesis of chiral secondary fatty alcohols from renewable unsaturated fatty acids. Two sequential steps involving fatty acid hydratase and fatty acid hydroxylating enzymes were optimized respectively, providing satisfied results for the model reaction. Moreover the substrate scope of the current system is also fairly broad giving access to enantiomerically pure alcohols. This photobiocatalytic synthesis route represents a more practical and environmentally less demanding alternative for the common chemical synthesis of long chain secondary alcohols with highly optical purity. This exciting work is therefore of significant interests to organic chemists and biotechnology engineers and is suitable for publication in Nature Communications with

Thank you very much for your positive evaluation and your helpful comments to improve the manuscript!

minor revisions.

1. References should be needed in L86-87, P4, for the pH property of CvFAP and LrOH, or some supplementary data of their pH property. Following the referee's suggestion, appropriate references have been added to the revised manuscript.
2. It is very interesting to use triolein as the starting material and involve three enzymes in this one pot as shown in Scheme 2. It would be helpful for readers to know clearly the whole reaction process and the changing concentrations of some main intermediate during the reaction time course. We fully agree with the referee that the reaction starting from the triglyceride is indeed future-pointing especially considering the production of 9-heptadecane as building block for organic photosensitisers from renewable starting materials. Unfortunately, our current personnel situation does not allow to investigate this reaction (such as the time course and an optimisation of the reaction conditions) in more detail. We sincerely hope, that the referee will understand this situation. We have, however, added her/his suggestion to the revised manuscript. We also included the product concentration in Table S1.
3. References should be needed in L122-123, P6, for the reported stereospecificity of oleate hydratases. Corrected as suggested by the reviewer. We added a reference that confirms the stereoselectivity.
4. The number of Figure shown in L124, P7 is wrong. It should be Figure 2. I suggest the author to add compound numbers for all products or substrates. Thus it will be easy for readers to catch the compound structure, for example in L131, P7, linoleic acid and so on. We have now revised the manuscript thoroughly and all compounds have been added with a number.
5. Moreover, it would be better to add some discussion about the effect of the substrate structures on the reaction results shown in Figure 1 (P7). Concerning the determination of the ee values of these products, I suggest the author to add some explanation in the text for the reason why NMR of the derivatization of (S)-(+)-O-acetylmandelic acid was used instead of chiral GC or HPLC. Due to the fact that many of the fatty alcohol products are novel (i.e. all unsaturated fatty alcohols obtained in this work were not reported in previous publications.), there is a lack of reference and therefore it is challenging to use chiral GC/HPLC, therefore, we used the ¹H NMR method. We have now included an explanation in the manuscript.
6. In the Authors contributions, XX and YJP did not appear in the author list. Corrected as suggested by the reviewer.
7. In Supplementary Information, L273, P12, one molecular formula of TMS should be reversed. In Figure S10, is palmitic acid as the starting substrate right? In Figure S16, the molecular formula is lost. Corrected as suggested by the reviewer.

Overall, the photobiocatalytic synthetic route of enantiomerically pure long chain secondary alcohols is original, the work is technically well-executed and the conclusions are firmly-supported by experimentation. It is therefore suitable for publication in Nature Communications after these comments are addressed.

Thank you very much for sharing our enthusiasm on this novel catalysis concept!

Reviewer #3

Zhang and colleagues report the use of tandem reactions comprised of two enzyme catalysts (hydratases and flavin-dependent decarboxylase) to facilitate the conversion of oleic acid to secondary alcohols as mono or diols. As the authors state, the transformation could potentially be of interest as an alternative and renewable means to access the synthesis of stereo-specific alcohols.

The manuscript is primarily comprised of turnover data and some reaction modification (pH and solvent compatibility). Although the majority of work is generally fine, I do not believe it to represent a major advance, nor is the significance sufficient enough for publication in Nature Communications. Ultimately, the reported substrate/product scope is relatively narrow, the products reported are not particularly interesting/valuable, and there is no fundamentally new enzymology established for the enzymes in question.

Additionally, as the manuscript is written, it is unclear if the authors are really achieving the significant TTONs that would enable a new synthetic scheme that is more advantageous than current methods

The turnover numbers observed in this study were in the range of 10.000, which according to the evaluation by Woody and coworkers (now cited in the revised manuscript) is at least sufficient for the synthesis of fine chemicals. Obviously, this number needs to increase by at least two orders of magnitude to attain feasibility for bulk chemicals! However, we would like to mention that we have not determined total turnover numbers (i.e. we have not yet tested the lifetime limits of both enzymes).

Although the characterization of the products is generally sound (relying on MS and NMR in some cases), the absence of any raw GC data prohibits the description of any potential side or intermediate products, TTONs, and accurate quantitation. Thus, there are several claims in the manuscript with regards to the performance of one or both enzymes as a function of substrate, pH, etc., that are impossible to substantiate.

The raw GC data, including the control reactions, have now been provided in the revised manuscript (in Supplementary Information).

The authors should provide raw GC chromatograms of reactants and products to show conversion of each species to validate several of these claims.

We have added raw GC data in the revised manuscript. However, the chromatograms are not as “clean” as we dreamed of. This is due to the fact that, after the silylation process, some unknown and very reproducible peaks (5.980, 7.382, 8.391, 8.772, 10.16, 10.220, 11.019, 12.34 min, respectively) appeared in GC chromatogram. A careful investigation only using substrates confirmed that this phenomenon is due to the artifacts in trimethylsilyl derivatization reactions. This is frequently observed in literatures (e.g. [https://doi.org/10.1016/S0021-9673\(99\)00267-8](https://doi.org/10.1016/S0021-9673(99)00267-8)). Pleasantly, these peaks are not overlapping with the substrates, products, and intermediates. Therefore, we decided to continue with the current derivatisation method for quantification. On the other hand, due to the lack of reference compounds, in the study of substrate scope we focused on the characterization and quantification of the final fatty alcohol products.

A minor point is that some of the authors’ product quantitation relies on assumed rather than empirically determined peak response relationships, which would limit accuracy of the reported conversion yields.

We fully understood the reviewer’s concern. Due to the fact that many of the fatty alcohol products are novel (i.e. all unsaturated fatty alcohols obtained in this work were not reported anywhere in previous publications), there is a lack of reference for the quantification. Therefore we had to compromise, we used 9-heptadecanol that has similar physiochemical properties as the other products for semi-quantification. That’s the original reason we called it “semi-quantification”.

Although this work is of some interest, I do not believe that it meets the high bar required for publication of Nature Communications and would be better suited to another journal.

Major points:

It is unclear whether any of the transformations under examination reach completion. What is the yield of the final product. TTON? Side products? No raw GC traces are shown, making it impossible to trace this information with the

Following the reviewer’s suggestion we have clarified the biotransformation descriptions including yields and side products. Raw GC data has been included in the revised supporting information.

protocol as given in the SI as the stock concentrations are not indicated.

Line 15 “novelhydratase” Why is this novel?
See next comment
Line 62: Rationale or literature precedent for this choice?

Following the reviewer’s suggestion we have clarified the (first) choice of the hydratase. The assumption of poor stability of the hydratase was based on the time courses, which has been clarified in the revised manuscript.

Line 67: Control data should be shown in the SI

Changed as suggested by the reviewer.

Line 70: “attribute this to a relative poor stability of *LrOH* under these conditions”
Literature precedent or a control reaction? Needs better explanation.

Following the reviewer’s suggestion we have clarified the (first) choice of the hydratase. The assumption of poor stability of the hydratase was based on the time courses, which has been clarified in the revised manuscript.

Line 73: stating

Changed as suggested by the reviewer.

Line 74: I do not understand the nomenclature with @. It seems awkward

Changed as suggested by the reviewer.

Line 76: “predominantly yielded the decarboxylation product of oleic acid” This implies that *LrOH* cannot dehydrate an alkene. Is there precedent in the hydratase literature? This should be tested directly to demonstrate the substrate scope of *LrOH*

The reviewer is absolutely right. The wild-type enzymes known so far exclusively convert carboxylic acids. This has been clarified in the revised manuscript.

Line 81: “considerably faster....”

Changed as suggested by the reviewer.

Line 85 “pH has a significant influenceboth enzymes” Unclear from the figure as to whether the pH is affecting one or both enzymes as only the final product (or consumption of the initial substrate) is being followed here.

In view of the minor importance of this aspect for the proof-of-concept reported in this manuscript, we have moved this part into the supporting information. Further studies will focus on the complete characterisation of the individual enzymes.

Line 99 “determination of hydratase activitynot straight forward (sic)” vague. I do not know what the authors are referring to.

Changed as suggested by the reviewer.

Figure 1 legend: What does the quantity for @*E. coli* refer to? *LrOH*? Total protein? The text indicates that this is total “lyophilized

Changed as suggested by the reviewer.

cells". However, this implies otherwise

Line 104 "substrate reservoir...product sink" Is this known that the alcohol products will partition in the organic layer as shown? It would also be worth comparing inclusion/omission of the cosolvent system to determine how/if this enables better conversion.

Changed as suggested by the reviewer. We clarified this point by referring to previous studies using the two-liquid phase approach.

Line 108 "quasi-irreversible hydratation" Why quasi?

This statement of is minor importance and therefore was removed.

Line 110 – "acidification of aqueous layer" as the authors have a simple solution in mind, it would be rather easy to test in a systematic fashion

We fully agree with the reviewer that this can be investigated rather easily in a systematic fashion. However, we hope that the reviewer will understand that this should be a topic of a future follow-up study. We have added an appropriate statement to the revised manuscript.

Line 111-114 Neither the acidification of the aqueous layer nor the "decarboxylation slowed down" are shown. What products are observed in this setup?

Changed as suggested by the referee.

Line 119 there is now Figure 2, but rather two Figure 1.

Corrected as suggested by the reviewer.

Line 123 "in line with the reported..." a reference is needed. Is this particular ortholog known to be stereospecific? Any other lines of evidence?

Corrected as suggested by the reviewer.

137 – "Especially 5,8-diol synthase from ...caught our attention" I do not really understand the impact of this choice. Is (Z)-heptadec-8-ene-4,7-diol particularly valuable?

Our motivation was purely curiosity about this unusual reaction (introducing two hydroxyl groups in one step). From our point of view, this sufficiently rationalises the choice of this interesting enzyme. We have just initiated collaboration with microbiology groups to investigate the biological activity of these compounds.

Line 145 "which we attribute" Completely unsubstantiated comment regarding the cytotoxicity. There are a myriad of other potential reasons the authors may wish to consider.

The reviewer is absolutely right with her/his comment here. We have deleted this statement.

Line 162, Figure 3: These products should be shown. Where does the hydroxyperoxide derive from? The synthetic reaction was performed for “m h”?	Corrected as suggested by the reviewer.
SI line 213 “The derivatized 213 fatty alcohols and 9-heptadecanol were assumed to have the same GC response factor.” This is very unlikely to be correct and casts some doubts as to the reported yields.	We fully understood the reviewer’s concern. Due to the fact that many of the fatty alcohol products are novel (i.e. all unsaturated fatty alcohols obtained in this work were not reported anywhere in previous publications.), there is a lack of reference for the quantification. Therefore we had to compromise, we used 9-heptadecanol that has similar physiochemical properties as the other products for semi-quantification. That’s the origin reason we called it “semi-quantification”. We hope that the reviewer could understand the circumstances.
Minor points Overall, I found there to be significant problems with the accuracy and quality of the writing. I would also give some thought as to what constitutes a paragraph, and several instances of absent/incorrect citation.	Following the reviewer’s suggestion we have done our best to improve the quality of the manuscript.
line 24: improper use of a semicolon	Corrected as suggested by the reviewer.
line 40: verb incorrect tense	Corrected as suggested by the reviewer.
line 40: “broadening its substrate scope” I don’t understand how the enzyme substrate scope has been significantly broadened in the present work beyond the addition of an alcohol.	Corrected as suggested by the reviewer.
line 69: incorrect use of colon	Corrected as suggested by the reviewer.
Line 105: “See SI” Where exactly?	Corrections have been made.

Reviewers' comments:

Reviewer #1 (Remarks to the Author):

I have to say that the revised manuscript has indeed shown some significant improvements by addressing a majority of concerns, suggestions, and criticisms from the three reviewers. However, there remain two crucial issues that prevent me from providing a recommendation of acceptance for this work. 1. As the Reviewer #3 pointed out, the products are very unlikely to have the same GC response factors as that of 9-heptadecanol. Thus, the product quantification by this method is unrealistic. The compromise by simply calling it "semi-quantification" is unacceptable. The products with similar physiochemical properties could show distinct GC response factors, which would make the reported yields problematic. 2. The value difference between the starting materials and products seems not to be large. Considering the suboptimal yields, the costs on product purification would be significant, casting some doubts on economic feasibility of this strategy. One minor issue: There are no page and line numbers in the column of authors' responses, which causes big difficulties for the reviewers to fast track the changes.

Reviewer #2 (Remarks to the Author):

The authors have adequately addressed all my comments through their revisions to the text, and have made a number of other improvements based on the advice of the other reviewers. I strongly support the publication of this work.

Reviewer #3 (Remarks to the Author):

The authors have very thoroughly addressed my concerns with the original manuscript. In particular, they have added more thorough characterization of the reaction products via the addition of raw chromatograms as well as more accurate quantitation methods. I feel that this manuscript is now of suitable quality for publication.

Again, we thank the reviewers for their ongoing efforts. We are very pleased that Reviewers #2 and #3 are satisfied with the changes made to the original manuscript and their positive evaluation of the revised manuscript.

Reviewer #1 suggested to re-investigate the quantification of the products as some doubts remained about the accuracy of the GC-based quantification. We have performed additional experiments quantifying the reagents via established NMR methods. As detailed below, these new results largely confirm the quantification reported in the previous version.

All changes made to the previous version have been marked in yellow.

Reviewer #1	Answer																								
I have to say that the revised manuscript has indeed shown some significant improvements by addressing a majority of concerns, suggestions, and criticisms from the three reviewers. However, there remain two crucial issues that prevent me from providing a recommendation of acceptance for this work.	Thank you very much for your overall positive evaluation. As outlined below, we have done our best to address the last remaining issues.																								
1. As the Reviewer #3 pointed out, the products are very unlikely to have the same GC response factors as that of 9-heptadecanol. Thus, the product quantification by this method is unrealistic. The compromise by simply calling it "semi-quantification" is unacceptable. The products with similar physiochemical properties could show distinct GC response factors, which would make the reported yields problematic.	Following the reviewer's suggestion we have performed the transformations shown in Figure 2 again and have quantified the concentrations of the reagents via ¹ H NMR spectroscopy using internal standards. The conversions determined are shown in the new Tables S2 and S3. As shown below, both methods correlated well. The difference for compound 8c may also be attributed to the fact that for the GC and the NMR method two individual experiments have been performed. We have added a comment on the alternative quantification methods to the caption of Figure 2.																								
	CpdConversion [%]NMRGC3c31244c65665c75746c43567c44328c4620	Cpd	Conversion [%]			NMR	GC	3c	31	24	4c	65	66	5c	75	74	6c	43	56	7c	44	32	8c	46	20
Cpd	Conversion [%]																								
	NMR	GC																							
3c	31	24																							
4c	65	66																							
5c	75	74																							
6c	43	56																							
7c	44	32																							
8c	46	20																							

2. The value difference between the starting materials and products seems not to be large. Considering the suboptimal yields, the costs on product purification would be significant, casting some doubts on economic feasibility of this strategy.

We fully agree with the reviewer that the method as described in this proof-of-concept study is not directly applicable to industrial implementation. However, considering that both the hydratase reaction (DOI: 10.1016/j.jbiotec.2012.01.002 and 10.1002/biot.201500141) and the photodecarboxylation reaction (DOI: 10.1002/cptc.201900205) have been performed with higher product titres (>50 mM), we are confident, that further optimisation will result in more practical reaction schemes.
A note acknowledging this has been added to the revised manuscript (last paragraph).

One minor issue: There are no page and line numbers in the column of authors' responses, which causes big difficulties for the reviewers to fast track the changes.

We apologise for the inconvenience this has caused. We hope that the localisation of the changes made to the revised manuscript is more straightforward this time.

REVIEWERS' COMMENTS:

Reviewer #1 (Remarks to the Author):

I think all of my concerns and suggestions have been appropriately addressed. I would suggest "Accept as is".